# Assessment of circulating microRNA specific for patients with familial adenomatous polyposis

Tomoki Yamano[1]*, Shuji Kubo[2], Emiko Sonoda[2], Tomoko Kominato[1], Kei Kimura[1], Michiko Yasuhara[1], Kozo Kataoka[1], Jihyung Son[1], Akihito Babaya[1], Yuya Takenaka[1], Takaaki Matsubara[1], Naohito Beppu[1], Masataka Ikeda[1]

1 Division of Lower GI Surgery, Department of Surgery, Hyogo College of Medicine, Hyogo, Japan,
2 Laboratory of Molecular and Genetic Therapeutics, Institute for Advanced Medical Sciences, Hyogo College of Medicine, Hyogo, Japan

* yamanot@hyo-med.ac.jp

## Abstract

Circulating microRNAs (miRNAs) are considered promising biomarkers for diagnosis, prognosis, and treatment efficacy of diseases. However, usefulness of circulating miRNAs as biomarkers for hereditary gastrointestinal diseases have not been confirmed yet. We explored circulating miRNAs specific for patients with familial adenomatous polyposis (FAP) as a representative hereditary gastrointestinal disease. Next-generation sequencing (NGS) indicated that plasma miR-143-3p, miR-183-5p, and miR-885-5p were candidate biomarkers for five FAP patients compared to three healthy donors due to moderate copy number and significant difference. MiR-16-5p was considered as an internal control due to minimum difference in expression across FAP patients and healthy donors. Validation studies by real-time PCR showed that mean ratios of maximum expression and minimum expression were 2.2 for miR-143-3p/miR-16-5p, 3.4 for miR-143-3p/miR-103a-3p, 5.1 for miR-183-5p/miR-16-5p, and 4.9 for miR-885-5p/miR-16-5p by using the samples collected at different time points of eight FAP patients. MiR-143-3p/16-5p was further assessed using specimens from 16 FAP patients and 7 healthy donors. MiR-143-3p was upregulated in FAP patients compared to healthy donors ($P = 0.04$), but not significantly influenced by clinicopathological features. However, miR-143-3p expression in colonic tumors was rare for upregulation, although there was a significant difference by existence of desmoid tumors. MiR-143-3p transfection significantly inhibited colorectal cancer cell proliferation compared to control microRNA transfection. Our data suggested regulation of miR-143-3p expression differed by samples (plasma or colonic tumors) in most FAP patients. Upregulation of plasma miR-143-3p expression may be helpful for diagnosis of FAP, although suppressive effect on tumorigenesis seemed insufficient in FAP patients.

## Introduction

MicroRNAs (miRNAs) are non-coding RNAs that regulate multiple biological processes in diseases through the post-transcriptional regulation of gene expression, including in cancer,

**Data Availability Statement:** All relevant data are within the manuscript and its Supporting Information files.

**Funding:** TY received grant from the Japanese Society for the Promotion of Science (JSPS) KAKENHI. Grant Number: 15K10154 URL: https://www-shinsei.jsps.go.jp/kaken/index.html The funders had no role in study design, data collection and analysis, decision to publish, or preparation of the manuscript.

**Competing interests:** The authors except for TY declare no potential conflicts of interest. TY currently has part-time engagement with Shionogi & CO., LTD. This work is not associated with this engagement. This does not alter our adherence to PLOS ONE policies on sharing data and materials.

**Abbreviations:** APC, adenomatous polyposis coli; CRC, colorectal cancer; cDNA, complementary DNA; FAP, familial adenomatous polyposis; miRNA, microRNA; NGS, next-generation sequencing; QC, quality control; RT-qPCR, reverse transcription quantitative PCR; TMM, trimmed mean of M-values.

cardiovascular disease, and neurodegenerative disease [1, 2]. Circulating miRNAs are recognized as diagnostic or prognostic biomarkers of many diseases [3]; however, their usefulness in the diagnosis of gastrointestinal hereditary diseases has not been reported [4–6]. Genetic variants in patients with hereditary disease may induce consistent and specific miRNA profile differences in their blood compared to healthy donors.

Familial adenomatous polyposis (FAP) is a representative inherited disease caused by germline mutations in the adenomatous polyposis coli (*APC*) tumor suppressor gene, which results in hundreds to thousands of colorectal adenomatous polyps [7–9]. We have experienced hundreds of FAP patients for several decades [10]. Without any treatment for polyposis, FAP patients over several decades may develop colorectal cancer (CRC) [11]. Thus, *APC* mutations are also essential for CRC progression in most cases of sporadic CRC [12]. The assessment of miRNAs in CRC patients might involve the miRNAs of FAP patients. However, we hypothesized that plasma miRNA profile of FAP patients, which bears *APC* variants in all cells, may differ from that of patients with sporadic CRC and healthy donors. Although most FAP cases are diagnosed by endoscopic examination without further genetic test, application of endoscopic examination for younger family is usually difficult to perform. Lynch syndrome, another hereditary disease, is difficult for diagnosis by endoscopic examination alone, and needs further examinations including genetic test [13]. Therefore, we chose FAP as a representative hereditary disease to evaluate the possibility of circulating miRNA as a potential biomarker of hereditary gastrointestinal diseases.

In this study, we explored the candidate plasma mir-RNAs specific for FAP patients by Next-generation sequencing (NGS) at first and validated the candidate miRNA by further experiments using clinical samples. We also assessed the effect of the candidate miRNA on proliferation of colorectal cancer cells.

## Material and methods

### Patients and samples

This study was performed in accordance with the guidelines and approval by the Institutional Review Board of Hyogo College of Medicine, Hyogo, Japan (Protocol No. 0288). Written informed consent was obtained from all patients prior to specimen collection in accordance with the Declaration of Helsinki. Ethylenediaminetetraacetic acid containing plasma samples were collected from 18 patients who were diagnosed as FAP by clinical features including family history and endoscopic findings in not only colon and rectum but also stomach and duodenum. Genetic tests for diagnosis of FAP have not been performed in this study, because these tests were not covered by insurance in Japan. These patients were treated at our hospital between 2015 and 2018. In addition, plasma samples were also collected from seven healthy donors. Plasma samples from the FAP patients were repeatedly collected, regardless of whether they received a total proctocolectomy. Experimental procedures applied to the FAP patients are listed in Table 1. Blood samples were centrifuged at $3,000 \times g$ for five min and the plasma was collected and stored at −80˚C until use.

Colonic tumor specimens of six of the FAP patients for which plasma samples were collected, in addition to another 13 FAP patients for which plasma samples were not collected, were assessed for miR-143-3p expression. The tumor specimens were placed in RNAlater (Thermo Fisher, Tokyo, Japan) at the time of collection and stored at −80˚C until use. Details on these patients are shown in Table 1.

### Next-Generation Sequencing (NGS)

Extraction of miRNA from plasma sample, library construction, and miRNA sequencing were performed by Exiqon Services (Vedbek, Denmark) as previously detailed [14]. Briefly, RNA

**Table 1. Patients with Familial Adenomatous Polyposis (FAP) used for plasma microRNA analysis in the current study.**

| Sample | Age at sampling (y) | sex | NGS[a] | Fluctuation | Validation | Pathology of colonic tumor | Gastric polyp | Duodenal polyp | Desmoid | Family history of FAP |
|---|---|---|---|---|---|---|---|---|---|---|
| F1 | 25 | M | | before[b]/ after[c] | before | T3 | (+) | (+) | (−) | (+) |
| F2 | 30 | F | before | after | after | Tis | (+) | (+) | (−) | (+) |
| F3 | 63 | M | | after | after | Tis | (+) | (+) | (−) | (+) |
| F4 | 29 | F | | before | before | adenoma | (+) | (+) | intra-abdominal | (+) |
| F5 | 45 | F | | after | after | adenoma | (+) | (+) | (−) | (−) |
| F6 | 22 | M | before | | after | adenoma | (+) | (−) | (−) | (−) |
| F7 | 41 | F | | | after | adenoma | (+) | (−) | (−) | (+) |
| F8 | 33 | M | before | after | after | adenoma | (+) | (+) | (−) | (+) |
| F9 | 19 | F | | | before | adenoma | (+) | (+) | abdominal wall | (−) |
| F10 | 24 | F | before | | | T1 | (+) | (−) | (−) | (−) |
| F11 | 43 | F | | | after | T3 | (+) | (+) | intra-abdominal | (+) |
| F12 | 56 | F | before | | | adenoma | (+) | (+) | (−) | (+) |
| F13 | 46 | F | | after | after | T3 | (+) | (+) | (−) | (−) |
| F14 | 35 | F | | | before | T3 | (−) | (−) | (−) | (−) |
| F15 | 26 | M | | before/after | before | T1 | (+) | (+) | chest wall | (−) |
| F16 | 31 | M | | | before | adenoma | (+) | (+) | chest wall | (−) |
| F17 | 43 | M | | | before | T3 | unknown | unknown | (−) | (−) |
| F18 | 31 | F | | | after | adenoma | (+) | (+) | intra-abdominal | (−) |

[a]Next-generation sequencing

[b]before surgery

[c]after surgery.

was extracted from 500 μl of plasma from each patient using a proprietary RNA isolation protocol optimized for plasma without the use of a carrier. The total RNA was eluted into ultra-low volumes.

The library preparation was performed using a NEBNEXT Small RNA Library Preparation Kit (New England Biolabs, Ipswich, MA, USA). A total of 6 μl of total RNA was used for generating the miRNA NGS libraries. RNA was converted to complementary DNA (cDNA) after ligation of adapters. The cDNA was amplified by 18 cycles of PCR. Library preparation quality control (QC) was performed using either Agilent Bioanalyzer 2100 or Agilent TapeStation 4200 system (Agilent Technologies, Lexington, MA, USA). Based on the quality of the inserts and concentration measurements, the libraries were pooled at equimolar ratios. The fraction corresponding to the miRNA libraries (< 145 nucleotides) was selected using a KAPA Library Quantification Kit (KAPA Biosystems, Wilmington, MA, USA). The library pools were then sequenced on a NextSeq 500 System (Illumina, San Diego, CA, USA), according to the manufacturer instructions. The raw data for each sample was de-multiplexed and FASTQ files were generated using the FastQC tool. FASTQ file quality scores used quality score binning and enabled a more compact storage of the raw sequences. Using eight levels of quality (0, 6, 15, 22, 27, 33, 37, 40), the method was tested and shown to result in virtually no loss of information [14].

Sequencing reads were mapped to a reference genome and the miRNA reads were mapped to miRBase release 20 (www.mirbase.org). The trimmed mean of M-values (TMM) normalization method was used for statistical analyses. Differential expression analyses were performed using TMM in the EdgeR statistical software package (Bioconductor http://www.bioconductor.org/). Statistical comparisons between FAP patients and healthy controls were performed in the identified miRNAs.

## Reverse transcription quantitative PCR (RT-qPCR) analysis of plasma samples

Aliquots of plasma samples (200 μl) were subjected to miRNA extraction using a miRNeasy Serum/Plasma Advanced Kit (QIAGEN, Tokyo, Japan), according to the manufacturer's instructions. The miRNA was eluted in 30 μl of RNase-free water and stored at −80˚C until use. The miRNA (2 μl) was reverse transcribed into cDNA using a miRCURY LNA RT Kit (QIAGEN), according to manufacturer's instructions. The RT-qPCR was performed using Power SYBR Green PCR Master Mix (Thermo Fisher), 1 μl of miRNA cDNA as template, and specific PCR primer sets for target miRNAs (miR-143-3p, miR-885-5p, miR-183-5p) and internal control miRNAs (miR-16-5p and miR-103a-3p) (QIAGEN). The relative expression of the miRNAs in each sample was compared to that of a healthy donor and the quantity calculated using the $2^{-\Delta\Delta Ct}$ method. For eight patients with multiple samples, the ratio of miRNA levels at two different time points was assessed to identify the potential biomarker that exhibited the least amount of variation. The candidate miRNA was then further validated in the 16 FAP patients.

## RT-qPCR analysis of colonic tumor specimens

Total RNA was extracted from the colonic tumor specimens and adjacent normal mucosa using an RNeasy Mini Kit (Qiagen) and reverse transcribed using a TaqMan Reverse Transcription Kit (Qiagen). The real-time quantitative PCR was performed using TaqMan MicroRNA assay with primers and probe specific for miR-143-3p (Qiagen). Primers and probe specific for RNU6B were used as an internal control. The relative changes in expression of the miRNAs in the colonic tumors compared to that in the corresponding normal mucosa were calculated using the $2^{-\Delta\Delta Ct}$ method.

## Cell culture

Human CRC cell lines, Caco2, HCT116, LoVo, and RKO were kind gifts from Dr. Hirofumi Yamamoto (Osaka University Graduate School of Medicine, Osaka, Japan). These cells were authenticated by ATCC service. These cells except for RKO were cultured by Roswell Park Memorial Institute 1640 medium with 10% fetal bovine serum (FBS). RKO was cultured by Dulbecco's modified Eagle's medium with FBS. Culture media and FBS was purchased from Nacalai Tesque (Kyoto, Japan) and Atlas Biologicals (CO, USA), respectively. Cell experiments were maintained at 37˚C in a humidified atmosphere of 5% $CO_2$.

## Cell transfection

Cells were seeded in 96-well flat-bottom plate at a density of 5000 cells/well for Caco-2, 500 cells/well for HCT116, 1000 cells/well for LoVo, and 500 cells/well for RKO, respectively. 24 hours after seeding, cells were treated with Lipofectamine 2000 and Opti-MEM containing 5 pmol of miRNA Mimic Negative Control (Control treatment), 5 pmol of hsa-miR-143-3p (miR-143 treatment), and no miRNA (Mock treatment) according to the manufacturer's

instructions. Transfection reagents including miRNAs were purchased from Thermo Fisher Scientific.

## Cell proliferation assay

In vitro growth was evaluated using alamarBlue Cell Viability Reagent at one, two, three, and four days after transfection. The ration of the fluorescence intensity at 590 nm by treatment groups compared to Mock treatment was calculated. Effect of miR-143-3p on cell growth was evaluated by comparison of the ratio between miR-143 treatment and Control treatment using Student's t-test.

## Statistical analysis

Statistical analysis was performed using JMP version 11 (SAS Japan, Tokyo, Japan). The influence of clinicopathological features on miRNAs of plasma and colonic tumor was assessed by Student's t-test when two groups were analyzed and analysis of variance when more than two groups were analyzed. The proportion of category was assessed by chi-square test. Comparison of effect on cell proliferation between miR-143 treatment and Control treatment was performed by Student's t-test at one, two, three, four days after transfection. $P < 0.05$ was considered significant for all analysis.

## Results

### Exploration of potential plasma biomarkers of FAP

Analysis of the NGS data revealed several miRNAs that were differentially expressed between normal controls and FAP patients (Fig 1A). Among the miRNAs identified, nine exhibited higher relative fold-changes and p-value and were considered potential candidates (Table 2). This included miR-143-5p, miR-96-5p, miR-218-5p, miR-106a-5p, miR-10b-3p, miR-143-3p, miR-885-5p, miR-183-5p, and miR-214-5p. Among these miRNAs in healthy donors, TMM

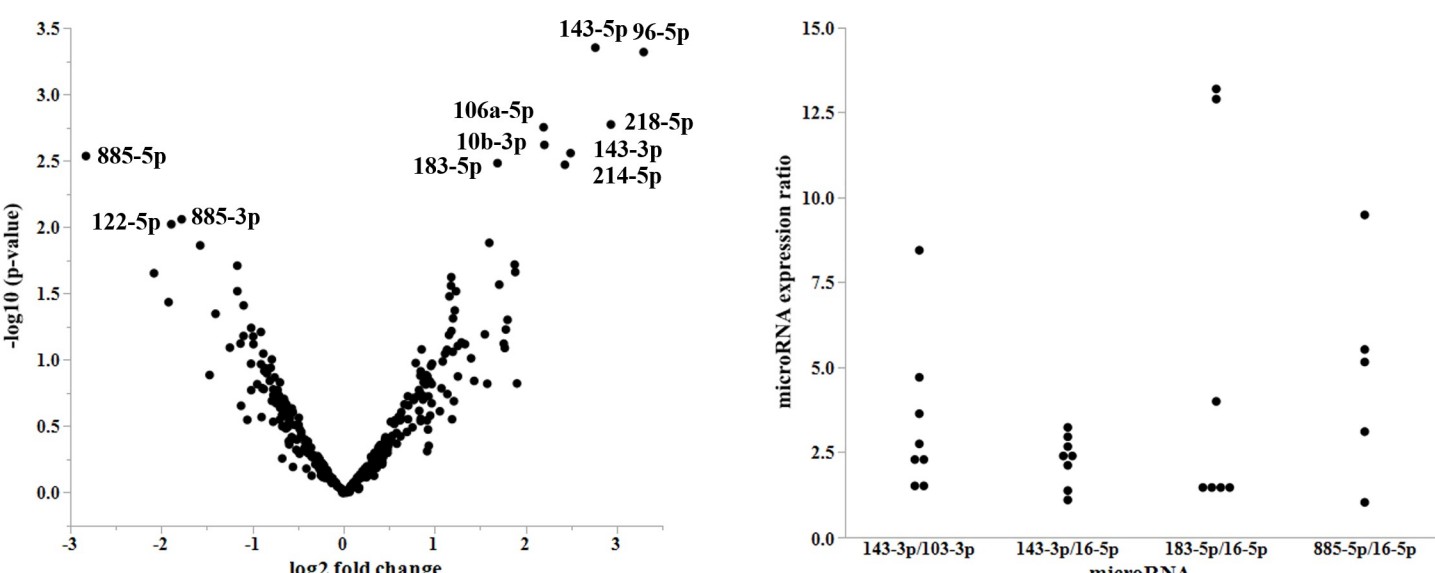

**Fig 1. A. Volcano plot of microRNAs (miRNAs).** This figure shows the expression levels of miRNAs in patients with familial adenomatous polyposis (FAP) compared to that in healthy donors. The differences are expressed as a log2 fold change on the X-axis and p-values on the Y-axis. B. Variation of microRNAs (miRNAs) expression levels. Expression ratios of the indicated miRNAs at different time points are plotted.

**Table 2. Candidate microRNAs detected by next-generation sequencing and results of Trimmed Mean of M-values (TMM) in samples.**

| MicroRNA | [a]Log2 (fold change) | P | Mean of patients | Mean of healthy donors | F2 | F6 | F8 | F10 | F12 | N1 | N2 | N3 |
|---|---|---|---|---|---|---|---|---|---|---|---|---|
| miR-143-5p | 2.76 | 0.0004 | 11.1 | 1.6 | 24.5 | 5.1 | 7.4 | 5.9 | 12.4 | 1.8 | 1.4 | 1.6 |
| miR-96-5p | 3.30 | 0.0005 | 10.9 | 1.2 | 7.6 | 20.3 | 9.1 | 6.3 | 11.4 | 0 | 0 | 3.6 |
| miR-218-5p | 2.94 | 0.002 | 35.1 | 4.5 | 102.7 | 10.6 | 6.8 | 15.4 | 40.0 | 6.2 | 5.8 | 1.6 |
| miR-106a-5p | 2.20 | 0.002 | 11.3 | 2.4 | 10.23 | 14.5 | 4.5 | 9.8 | 17.6 | 4.4 | 1.8 | 1.0 |
| miR-10b-3p | 2.21 | 0.002 | 20.0 | 4.4 | 21.0 | 27.4 | 12.5 | 27.6 | 11.7 | 0.7 | 4.7 | 7.8 |
| miR-143-3p | 2.49 | 0.003 | 23673 | 4208 | 75700 | 10410 | 6993 | 12593 | 12669 | 5486 | 2868 | 4269 |
| miR-885-5p | −2.83 | 0.003 | 2.1 | 15.0 | 0 | 7.4 | 0 | 3.2 | 0 | 12.4 | 13.4 | 19.2 |
| miR-183-5p | 1.69 | 0.003 | 1422 | 440 | 894 | 1521 | 985 | 1198 | 2510 | 322 | 420 | 579 |
| miR-214-5p | 2.43 | 0.003 | 29.2 | 5.4 | 65.3 | 20.3 | 4.0 | 20.1 | 36.2 | 9.5 | 4.0 | 2.6 |
| miR-885-3p | −1.78 | 0.009 | 7.09 | 24.4 | 2.4 | 16.0 | 6.2 | 4.7 | 6.0 | 17.6 | 7.9 | 47.7 |

[a]Log2 of fold change of TMM in mean of FAP patients (F2, F6, F8, F10, F12) compared to that of healthy donors (N1, N2, N3).

values varied from 1 to 4,208. Three miRNAs (miR-143-3p, miR-885-5p, and miR-183-5p) had more than 10 TMM values and were further analyzed for validation as a potential biomarker of FAP. NGS also showed that miR-16-5p was most stably expressed across FAP patients and healthy donors, which indicated a good candidate for normalizer in quantitative PCR study.

## Assessment of miRNA expression fluctuations

The ratios of minimum and maximum expression values at different time points were assessed to evaluate fluctuations of the levels of miRNAs in the samples. The mean ratios of miR-143-3p/miR-103a-3p, miR-143-3p/miR-16-5p, miR-183-5p/miR-16-5p, and miR-885-5p/miR-16-5p were 3.4, 2.2, 5.1, and 4.9, respectively (Fig 1B). As miR-143-3p exhibited the least expression difference relative to miR-16-5p, it was further assessed for validation as a potential biomarker of FAP.

## Validation

Specimens from 16 FAP patients and 7 healthy donors were assessed to validate the plasma findings (Table 1). Plasma levels of miR-143-3p were significantly upregulated in FAP patients compared to that in healthy donors ($P = 0.04$; Fig 2A). There was no significant difference in miR-143-3p expression based on pathological T factor ($P = 0.25$, Fig 2B), pathological node positive or not ($P = 0.42$), patient sex ($P = 0.29$), levels before or after radical surgery ($P = 0.43$), the existence of desmoid tumor ($P = 0.33$), or family history ($P = 0.91$).

## MiR-143-3p expression in colonic tumors

The expression of miR-143-3p in the colonic tumors was upregulated with fold change more than two in one patient alone (Table 3). Eight patients exhibited less than half expression in normal mucosa compared to the matched tumor samples. Then, upregulation was significantly rare by chi-square test ($P = 0.02$). There was no significant difference in miR-143-3p expression based on sex ($P = 0.55$), pathological T factor ($P = 0.78$, Fig 3), existence of gastric ($P = 0.14$) or duodenal polyp ($P = 0.59$), or family history ($P = 0.43$). However, there was a significant difference in miR-143-3p expression based on the existence or absence of desmoid tumor ($p < 0.0001$).

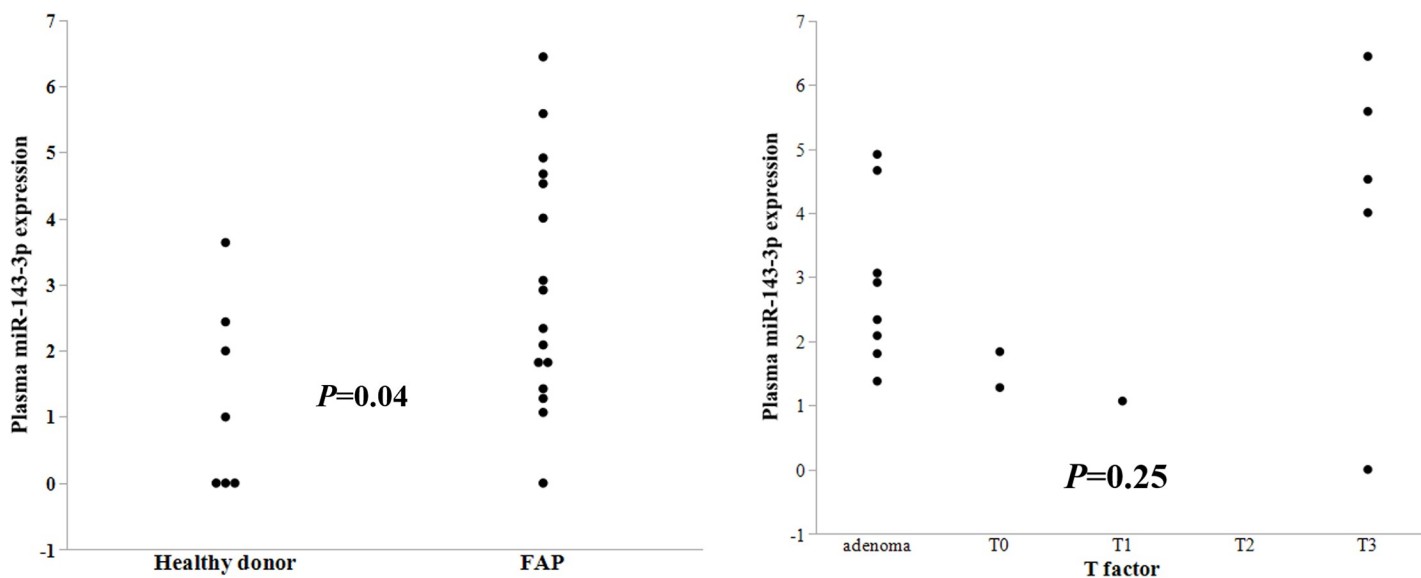

**Fig 2. A. Plasma miR-143-3p expression levels in patients with familial adenomatous polyposis (FAP) and healthy donors (Control).** There was a significant difference in miR-143-3p expression between the two groups ($P = 0.04$). B. Plasma miR-143-3p expression levels in patients with familial adenomatous polyposis (FAP) by T factor. There is no significant difference in miR-143-3p expression by T factor.

## Effect of miR-143-3p on the proliferation of colorectal cancer cells

We assessed the effect of miR-143-3p on the proliferation of CRC cell lines (Caco2, HCT116, LoVo, and RKO). Except for Caco2, miR-143 treatment showed significant growth inhibition on these cell lines compared to Control treatment at four days after transfection ($p < 0.01$, Fig

**Table 3. Relative miR-143-3p expression levels in colonic tumor.**

| Patient | [a]Relative miR-143-3p expression | Age (y) | Sex | Pathological T factor | Gastric polyp | Duodenal polyp | Desmoid | Family history of FAP |
|---------|-----------------------------------|---------|-----|------------------------|---------------|----------------|---------|------------------------|
| F1 | 0.55 | 25 | M | T3 | (+) | (+) | (-) | (+) |
| F6 | 0.30 | 22 | M | adenoma | (+) | (+) | (-) | (+) |
| F8 | 1.70 | 33 | M | adenoma | (+) | (+) | (-) | (+) |
| F11 | 1.94 | 41 | F | T3 | (-) | (+) | (+) | (+) |
| F16 | 2.92 | 31 | M | adenoma | (+) | (+) | (+) | (-) |
| F17 | 0.87 | 43 | M | T3 | Unknown | Unknown | (-) | (-) |
| F19 | 1.29 | 29 | F | adenoma | Unknown | Unknown | Unknown | (+) |
| F21 | 0.10 | 52 | F | T0 | (+) | (+) | (-) | (-) |
| F22 | 0.10 | 30 | M | T0 | (+) | (+) | (-) | (+) |
| F23 | 0.56 | 25 | F | adenoma | (+) | (+) | (-) | (+) |
| F24 | 0.49 | 13 | M | T0 | (+) | (+) | (-) | (+) |
| F25 | 0.08 | 19 | M | adenoma | (+) | (+) | (-) | (+) |
| F26 | 0.22 | 21 | F | adenoma | (+) | (-) | (-) | (+) |
| F27 | 0.66 | 32 | M | T3 | (+) | (+) | (-) | (-) |
| F28 | 1.08 | 69 | M | T4 | (-) | (-) | (-) | (-) |
| F29 | 0.25 | 24 | F | T1 | (+) | (-) | (-) | (-) |
| F30 | 0.95 | 39 | M | T2 | Unknown | Unknown | (-) | (-) |
| F31 | 0.19 | 25 | F | T3 | (+) | (+) | (-) | (+) |

[a]Relative miR-143-3p expression levels in colonic tumor compared to those in normal mucosa.

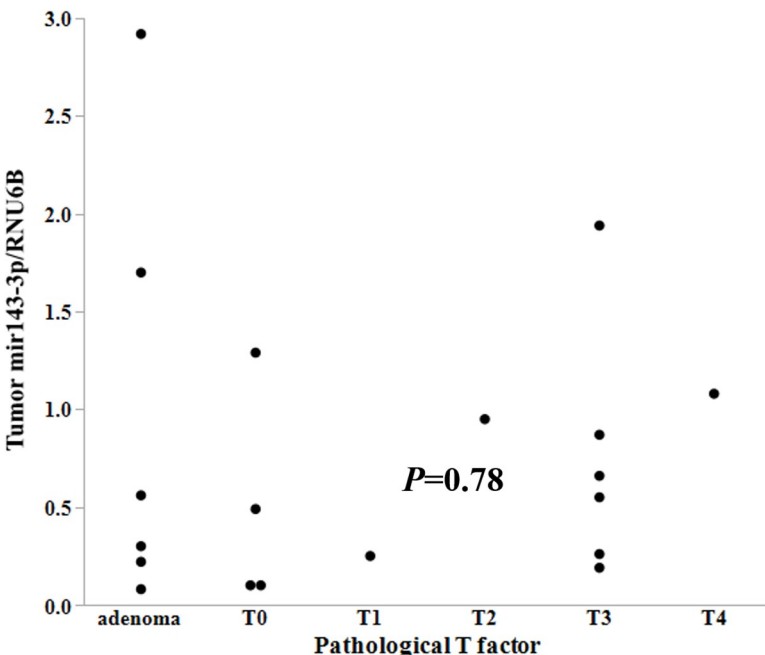

**Fig 3. MiR-143-3p expression in colonic tumors.** Expression levels of miR-143-3p relative to RNU6B are plotted versus pathological T factor. There was no significant difference in the expression of miR-143-3p relative to pathological T factor ($P = 0.78$).

4A–4D). MiR-143 treatment also showed significant inhibition of HCT116 and LoVo compared to Control treatment at three days after transfection ($p < 0.01$, Fig 4B and 4C). In Caco2, effect of miR-143-3p on proliferation was not significant at four days after transfection (Fig 4A, $P = 0.06$).

## Discussion

In the current study, we explored circulating miRNAs specific for FAP patients. We found that miR-143-3p was upregulated in plasma, but not in colonic tumors.

Circulating miRNAs are promising biomarkers for the diagnosis of CRC [15–17]. However, only a few reports have considered miR-143-3p as a biomarker of CRC [18–20]. On the other hand, circulating miR-143-3p has been upregulated in other diseases including acute ischemic stroke, cerebral atherosclerosis, amyotrophic lateral sclerosis, and pituitary adenoma [21–24].

MiR-143-3p expression in tumor has been reported to be downregulated in many malignancies including CRC, and the restoration of miR-143-3p expression inhibits the proliferation of malignancies in experimental models and is therefore considered a tumor suppressor miRNA [25–32]. Our data showed that transfection of miR-143-3p inhibited proliferation of CRC cell lines, although the extent of growth inhibition differed among these cell lines. Our results are similar to the previous reports, especially by Luo et al [26, 30, 32].

Although FAP patients are prototypic of CRC progression for tumors bearing *APC* mutations, previous assessments of miRNAs focusing on FAP patients have been limited due to rarity of FAP [6, 33]. Rarity of miR-143-3p upregulation in colonic tumors of FAP patients is consistent with a previous report [33].

Then, our current results seemed inconsistent with the previous studies that exogenous miR-143-3p could inhibit CRC tumor growth, because upregulation of plasma miR-143-3p

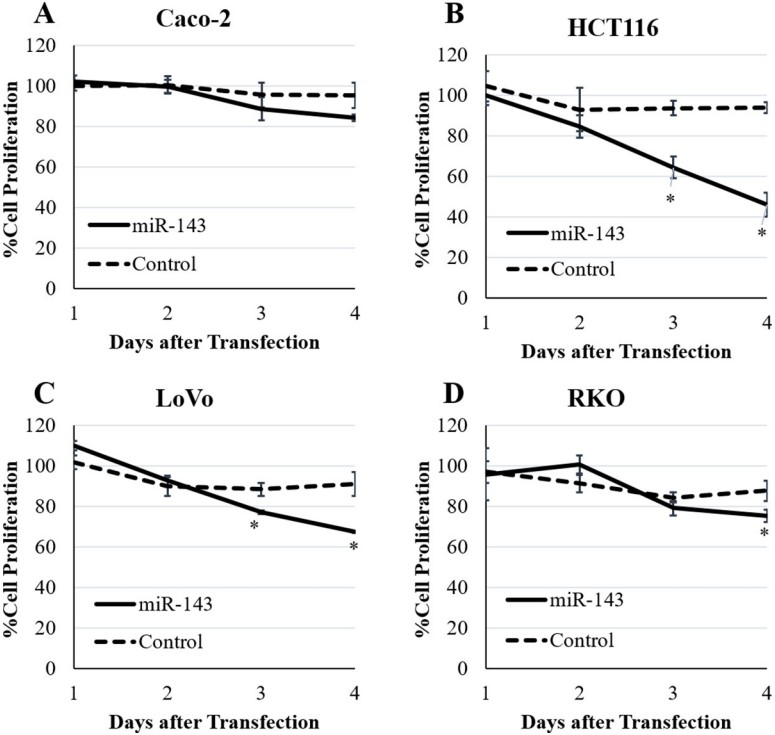

**Fig 4. The effect of miR-143-3p on the proliferation of colorectal cancer cells.** The effect of miR-143-3p on the cell growth was evaluated at one, two, three, and four days after transfection. There was no significant effect on proliferation of Caco2 (**A**). MiR-143-3p showed significant effect on proliferation of HCT116 (**B**) and LoVo (**C**) at three and four days after transfection ($p < 0.01$). MiR-143-3p showed significant effect on proliferation of RKO (**D**) at four days after transfection ($p < 0.01$). Results are expressed as mean of the ratio of cell proliferation compared to transfection without miRNA ± standard deviation. * $p < 0.01$ compared to Control treatment.

seemed insufficient for inhibiting tumor progression in FAP patients. However, transfection of miR-143-3p should be more effective on cell proliferation than plasma miR-143-3p.

There are many target genes of the tumor suppressor function of miR-143-3p in CRC, including *Bcl-2*, *CASP3*, *CD44*, *KLF5*, *KRAS*, *BRAF*, *TLR2*, *SOD1*, *IGF1R*, *ITGA6*, and *ASAP3* [26–32]. However, these results were performed using the cancer cell lines, but not benign tissues or cells from FAP patients or *APC* deficient mice.

Defective epithelial regeneration after injury is observed in miR-143/miR-145 deficient mice due to of stromal cell dysfunction, but not epithelial cell dysfunction [34]. Therefore, a constant upregulation of circulating miR-143-3p may be beneficial for epithelial regeneration in colonic tissues of FAP patients.

Our study had several limitations mainly because we have described the data from clinical samples, which were not supported by basic experiments.

First, biological effect of miR-143-3p have not been evaluated by using colonic samples derived from FAP patients or *APC*-deficient mice, although we have shown inhibitory effect of miR-143-3p on proliferation by using CRC cell lines.

Second, the sources of upregulated plasma miR-143-3p have not been confirmed. Possible sources of miR-143-3p could be mesenchymal cells such as fibroblast and smooth muscle cells [35]. Then, higher expression of miR-143-3p in colonic tumor with desmoid tumor (fibromatosis) in our study seemed to be associated with this speculation [36].

Third, we did not validate the usefulness of measuring circulating miR-143-3p in family members of FAP patients, who were not diagnosed with FAP or not.

Fourth, we used miR-16-5p as an internal standard based on NGS data since standard internal controls for plasma miRNAs have not been established yet. Therefore, our results will be validated by using miR-16-5p as an internal control.

## Conclusions

Plasma miR-143-3p was upregulated in FAP patients; however, miR-143-3p expression was rare for upregulation in colonic tumors, except for cases of desmoid tumors. Increased miR-143-3p expression seemed insufficient for prevention tumor progression in FAP patients. We expect that our findings will be validated in future studies.

## Supporting information

**S1 Checklist.**
(PDF)

**S1 Data.**
(XLSX)

**S2 Data.**
(XLSX)

## Acknowledgments

We thank Ms. Shino Tanaka, MS. Takako Ishikawa, and the members of Joint-Use Research facilities at the Hyogo College of Medicine for collecting data. We also thank Editage (www.editage.com) for English language editing.

## Author Contributions

**Conceptualization:** Tomoki Yamano.

**Data curation:** Tomoki Yamano, Shuji Kubo, Tomoko Kominato, Kei Kimura, Michiko Yasuhara, Kozo Kataoka, Jihyung Son, Akihito Babaya, Yuya Takenaka, Takaaki Matsubara, Naohito Beppu, Masataka Ikeda.

**Formal analysis:** Tomoki Yamano.

**Funding acquisition:** Tomoki Yamano.

**Investigation:** Tomoki Yamano, Emiko Sonoda, Tomoko Kominato.

**Methodology:** Tomoki Yamano.

**Project administration:** Tomoki Yamano.

**Resources:** Tomoki Yamano.

**Writing – original draft:** Tomoki Yamano.

**Writing – review & editing:** Tomoki Yamano, Shuji Kubo, Emiko Sonoda, Tomoko Kominato, Kei Kimura, Michiko Yasuhara, Kozo Kataoka, Jihyung Son, Akihito Babaya, Yuya Takenaka, Takaaki Matsubara, Naohito Beppu, Masataka Ikeda.

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
