## [Decision Letter · Decision Letter 0]

28 Jul 2020

PONE-D-20-13212

Assessment of circulating microRNA specific for patients with familial adenomatous polyposis

PLOS ONE

Dear Dr. Yamano,

Thank you for submitting your manuscript to PLOS ONE. After careful consideration, we feel that it has merit but does not fully meet PLOS ONE’s publication criteria as it currently stands. Therefore, we invite you to submit a revised version of the manuscript that addresses the points raised during the review process.

We look forward to receiving your revised manuscript.

Kind regards,

Sripathi M Sureban, Ph.D.

Academic Editor

PLOS ONE

Journal Requirements:

2. To comply with PLOS ONE submission guidelines, in your Methods section, please provide additional information regarding your statistical analyses. For more information on PLOS ONE's expectations for statistical reporting, please see https://journals.plos.org/plosone/s/submission-guidelines.#loc-statistical-reporting.

3.We note that you have indicated that data from this study are available upon request. PLOS only allows data to be available upon request if there are legal or ethical restrictions on sharing data publicly. For information on unacceptable data access restrictions, please see http://journals.plos.org/plosone/s/data-availability#loc-unacceptable-data-access-restrictions.

4. Your ethics statement must appear in the Methods section of your manuscript. If your ethics statement is written in any section besides the Methods, please move it to the Methods section and delete it from any other section. Please also ensure that your ethics statement is included in your manuscript, as the ethics section of your online submission will not be published alongside your manuscript.

5. Please upload a copy of Figure 3, to which you refer in your text on page 11. If the figure is no longer to be included as part of the submission please remove all reference to it within the text.

6. Please include a caption for figure 3.

Reviewers' comments:

Reviewer's Responses to Questions

**Comments to the Author**

1. Is the manuscript technically sound, and do the data support the conclusions?

Reviewer #1: No

Reviewer #2: Partly

2. Has the statistical analysis been performed appropriately and rigorously? 

Reviewer #1: No

Reviewer #2: Yes

3. Have the authors made all data underlying the findings in their manuscript fully available?

Reviewer #1: Yes

Reviewer #2: Yes

4. Is the manuscript presented in an intelligible fashion and written in standard English?

Reviewer #1: Yes

Reviewer #2: Yes

5. Review Comments to the Author

Reviewer #1: “Assessment of circulating microRNA specific for patients with familial adenomatous polyposis” by Yamano and colleagues is an effort to downselect a cohort of microRNAs from a group of clinical samples obtained from 15 familial adenomatous polyposis and 7 controls. Using next generation sequencing the authors identify miRs that may be potential biomarkers for pathology.

While the study is strengthened by the observation of clinical findings, the lack of any functional studies diminishes the enthusiasm in the study and the potential of the findings. In order for the study to be improved the following sets of data are highly encouraged to be introduced.

1. Gain and loss of functional stides usingselected microRNAs. Judicious choice of controls are warranted to strengthen the observations

2. What are the targets being regulated by the microRNAs of interests and how are they affecting the disease process?

3. Experimental evidence to highlight the biological/ physiological importance of occurrence of miRs in biosamples such as plasma and tumors- and also if there is an occurrence at 1 site, do they lose the potential to be biomarkers?

Reviewer #2: In this manuscript by Yamano et al, the authors are looking for circulating miRNA in patients with a hereditary disease called familial adenomatous polyposis. Both plasma and tissue miRNA expression was tested from patients with FAP vs control (tumor or mucosa for tissue).

1. The authors provide no rationale for performing the experiments. The expression of APC is used for FAP, and given that this is a hereditary disease, there is often family history present as well. The authors make no argument to the need of another marker for FAP. They describe in the introduction that they are going to compare 3 groups: healthy vs. FAP vs. CRC, and that potentially they could find a marker that would span both FAP and non-hereditary CRC. This wasn't done.

2. It's unclear why the authors use two control miRNAs for expression of miR143-3p, but not for any of the other potential candidates.

3. The Figures are mislabeled. There are only 2 figures (with A & B on both).

4. The expression of miRNA in the tumors is mostly decreased compared to control tissue, but there is no significant change. No explanation is provided for the difference between plasma and tumor tissue.

Overall, the major concern with this manuscript is the lack of rationale for the studies and the lack of description of the results.

6. PLOS authors have the option to publish the peer review history of their article (what does this mean?). If published, this will include your full peer review and any attached files.

Reviewer #1: No

Reviewer #2: No

---

## [Author Response · Author response to Decision Letter 0]

30 Aug 2020

Dear Reviewers,

Thank you for your time for reviewing the manuscript and describing the useful comments to us. We have revised the manuscript and responded to your comments as possible as we could.

We apologized to the reviewers for not performing the appropriate basic experiments to support our findings, although we agree the importance of the further experiments. We considered that new samples from the FAP patients or APC deficient mice were most suitable for the experiments. However, performing the experiments is beyond our capacity now. Then, we added some data to support our findings and mainly revised Abstract and Discussion to clearly describe the findings and problems of our study.

The association between plasma miR-143-3p and pathological T factor / lymph node metastasis and the rarity of mir-143-3p upregulation in colonic tumors were added in the revised manuscript.

Discussion was organized by circulating miR-143-3p, tumor miR-143-3p, inconsistent expression between plasma and tumor, target of mir-143-3p, speculation, and limitation of the study, because these contents were described roughly in the first version. 

Reviewer #1:

 “Assessment of circulating microRNA specific for patients with familial adenomatous polyposis” by Yamano and colleagues is an effort to downselect a cohort of microRNAs from a group of clinical samples obtained from 15 familial adenomatous polyposis and 7 controls. Using next generation sequencing the authors identify miRs that may be potential biomarkers for pathology.

While the study is strengthened by the observation of clinical findings, the lack of any functional studies diminishes the enthusiasm in the study and the potential of the findings. In order for the study to be improved the following sets of data are highly encouraged to be introduced.

1. Gain and loss of functional studies using selected microRNAs. Judicious choice of controls are warranted to strengthen the observations.

Thank you for your comment. Many studies using cancer cell lines have already confirmed onco-suppressive effect of mir-143-3p. Our data showed that circulating mir-143 was upregulated in FAP patients, although it seemed no significant influence on tumorigenicity by upregulated miR-143-3p. Then, we considered that basic studies using colonic samples from FAP patients or APC deficient mice may be more suitable for functional study than cancer cell lines. However, these experiments were beyond our capacity now. 

Instead of functional studies, we added the data regarding association between plasma mir-143-3p and pathological data (pathological T factor and lymph node metastasis) in Results. We also assessed tendency of miR-143-3p expression in colonic tumors and described in Results. We also revised Discussion.

Results

Validation

(First version) Line: 176-178

There was no significant difference in miR-143-3p expression based on patient sex (P = 0.29), levels before or after radical surgery (P = 0.43), the existence of desmoid tumor (P = 0.33), or family history (P =0.91).

(Revised version) Line200-204

There was no significant difference in miR-143-3p expression based on pathological T factor (P = 0.25, Fig. 2B), pathological node positive or not (P =0.42), patient sex (P = 0.29), levels before or after radical surgery (P = 0.43), the existence of desmoid tumor (P = 0.33), or family history (P =0.91).

MiR-143-3p expression in colonic tumors

(First version) No description.

(Revised version) Line:216-217

Then, upregulation was significantly rare by chi-square test (P =0.02).

Discussion

(First version) Line215-217

Our study had several limitations. First, our results regarding the upregulation of circulating miR-143 and downregulation of colonic tumor miR-143 has not been confirmed in studies using APC-deficient mice.

(Revised version) Line: 257-260

Our study had several limitations mainly because we have described the data from clinical samples, which were not supported by basic experiments. 

First, biological effect of miR-143-3p have not been evaluated by using colonic samples derived from FAP patients or APC-deficient mice. 

2. What are the targets being regulated by the microRNAs of interests and how are they affecting the disease process?

Thank you for your comment. Target genes of mir-143-3p against cancer cell lines have been reported already as we described in Discussion. In this study, upregulated mir-143-3p in FAP patients seemed ineffective to prevent tumorigenesis. Then, the specimens from FAP patients or APC deficient mice may be more preferable than cancer cell lines to assess the target genes. However, these experiments were beyond our capacity.

The source cells of upregulated miR-143-3p expression are also critical to explain our findings. We have added the sentence in revised Discussion as below.

Discussion

(First version) Line194-196

There are many target genes of the tumor suppressor function of miR-143 in CRC, including Bcl-2, CASP3, CD44, KLF5, KRAS, BRAF, TLR2, SOD1, IGF1R, ITGA6, and ASAP3 [14-19].

(Revised version) Line:249-252

There are many target genes of the tumor suppressor function of miR-143 in CRC, including Bcl-2, CASP3, CD44, KLF5, KRAS, BRAF, TLR2, SOD1, IGF1R, ITGA6, and ASAP3 [27-32]. However, these results were performed using the cancer cell lines, but not benign tissues or cells from FAP patients or APC deficient mice. 

3. Experimental evidence to highlight the biological/ physiological importance of occurrence of miRs in biosamples such as plasma and tumors- and also if there is an occurrence at 1 site, do they lose the potential to be biomarkers?

The reason why mir-143-3p showed different expression pattern by the location was undetermined. The critical issue should be the cellular source of plasma mir-143-3p, which is also elevated in the patients with cerebral atherosclerosis, acute ischemic stroke, and amyotrophic lateral sclerosis. Mesenchymal cells such as fibroblasts and smooth muscle cells have been reported to highly express mir-143-3p compared to colonic epithelial cells. Well-organized experiments are necessary for understanding the mechanisms of mir-143-3p in human.

We added the sentences in Discussion as below.

Discussion 

(First version) Line:196-197

However, our current results suggested the upregulation of plasma miR-143-3p was insufficient for inhibiting tumor progression in patients with FAP.

(Revised version) Line:245-248

Then, our current results were inconsistent with the previous studies because upregulation of plasma miR-143-3p seemed insufficient for inhibiting tumor progression in FAP patients. Difference of miR-143-3p expression by sample location and its interaction has not been evaluated. 

(First version) Line:212-214

Our finding that suggests that the upregulation of mir-143-3p in colonic tumor is associated with desmoid tumor should be validated by further studies. Desmoid tumor is considered a serious neoplasm related with FAP and difficult to diagnose [32].

(Revised version) Line: 261-264

Second, the sources of upregulated plasma miR-143-3p have not been confirmed. Possible sources of mir-143 could be mesenchymal cells such as fibroblast and smooth muscle cells [34]. Then, higher expression of miR-143-3p in colonic tumor with desmoid tumor (fibromatosis) in our study seemed to be associated with this speculation. [35]. 

Reviewer #2: 

In this manuscript by Yamano et al, the authors are looking for circulating miRNA in patients with a hereditary disease called familial adenomatous polyposis. Both plasma and tissue miRNA expression was tested from patients with FAP vs control (tumor or mucosa for tissue).

1. The authors provide no rationale for performing the experiments. The expression of APC is used for FAP, and given that this is a hereditary disease, there is often family history present as well. The authors make no argument to the need of another marker for FAP. They describe in the introduction that they are going to compare 3 groups: healthy vs. FAP vs. CRC, and that potentially they could find a marker that would span both FAP and non-hereditary CRC. This wasn't done.

We hypothesized that microRNA profiling in genetic diseases should be modified to compensate the disadvantages induced by genetic abnormality. Then, we considered that FAP patients should have the specific profiling of circulating microRNAs to compensate the influence by loss of APC function. We selected FAP as a representative genetic disease among gastrointestinal diseases, because we have constantly experienced FAP patients and published manuscripts for several decades. Another representative genetic disease, Lynch syndrome, has variable clinical and genetic features than FAP.

Most FAP patients are usually diagnosed by clinical features such as colonic polyposis and family history without confirming APC mutation. Diagnosis of Lynch syndrome is more difficult than that of FAP , and CRC patients are not routinely assessed for Lynch syndrome. We expected that effective diagnosis by circulating microRNAs for FAP could be applied for Lynch.

 We apologize that description is so insufficient that you misunderstand the study. The description showed our hypothesis that circulating microRNAs profiling maybe specific for FAP patients, which should be different from healthy patients and sporadic CRC patients. We have not assessed blood samples derived from sporadic CRC patients.

We have revised the sentences as below. And added reference 10 regarding FAP and reference 13 regarding Lynch syndrome.

Abstract 

(First version) Line: 34-36

We assessed circulating miRNAs of patients with familial adenomatous polyposis (FAP) as a hereditary disease and prototype of colorectal cancer progression.

(Revised version) Line:28-31

We assessed circulating miRNAs of patients with familial adenomatous polyposis (FAP) as a hereditary disease and prototype of colorectal cancer progression to evaluate as possible biomarkers of hereditary gastrointestinal diseases.

Introduction

(First version) No description

(Revised version) Line: 64-65

We have experienced hundreds of FAP patients for several decades [10].

(First version) Line:77-80

Therefore, we chose FAP as a representative hereditary disease and a prototype of CRC and attempted to identify and assess a plasma miRNA specific for patients with FAP as a potential biomarker.

(Revised version) Line:71-77

Although most FAP cases are diagnosed by endoscopic examination without further genetic test, application of endoscopic examination for younger family is usually difficult to perform. Lynch syndrome, another hereditary disease, is difficult for diagnosis by endoscopic examination alone, and needs further examinations including genetic test [13]. Therefore, we chose FAP as a representative hereditary disease to evaluate the possibility of circulating miRNA as a potential biomarker of hereditary gastrointestinal diseases.

Discussion

(First version) Line:189-191

In the current study, we explored circulating miRNAs specific for patients with FAP and found that miR-143-3p was upregulated in the plasma of patients with FAP, but not in colonic tumors.

(Revised version) Line:231

In the current study, we explored circulating miRNAs specific for FAP patients.

(First version) Line:217-218

Second, we did not validate the usefulness of measuring circulating miR-143 in family members of patients with FAP.

(Revised version) Line: 265-266

Third, we did not validate the usefulness of measuring circulating miR-143-3p in family members of FAP patients, who were not diagnosed with FAP or not. 

2. It's unclear why the authors use two control miRNAs for expression of miR143-3p, but not for any of the other potential candidates.

Different from tumor, control miRNA for blood samples has not been confirmed, although miR-103-3p is considered as one of the common control miRNAs. Next generation sequence data indicated miR-16-5p was suitable control miRNA, because of similar expression between FAP patients and healthy donors. Then, we assessed these two miRNAs. Mir-103a-3p is one of the internal control miRNAs of blood samples recommended by manufacture’s instruction from Exiqon.

Abstract

(First version) Line:39-40

With minimum differences among samples, miR-16-5p was selected as an internal control.

(Revised version) Line:34-35

MiR-16-5p was also assessed as an internal control due to minimum difference in expression across FAP patients and healthy donors. 

Results

Exploration of potential plasma biomarkers of FAP

(First version)

No description in Results.

(Revised version) Line:172-174

NGS also showed that miR-16-5p was most stably expressed across FAP patients and healthy donors, which indicated a good candidate for normalizer in quantitative PCR study. 

3. The Figures are mislabeled. There are only 2 figures (with A & B on both).

Thank you for your comment. We revised the figure numbers as below.

Results

Exploration of potential plasma biomarkers of FAP

(First version) Line:158

Fig.1

(Revised version) Line:166

Fig.1A

Assessment of miRNA expression fluctuations

(First version) Line:169

Fig. 2

(Revised version) Line:166

Fig.1B

Validation

(First version) Line: 175

Fig. 3

(Revised version) Line:200

Fig 2A 

(First version) No description

(Revised version) Line:202

Fig 2B

MiR-143-3p expression in colonic tumors

(First version) Line: 181

Fig. 2B

(Revised version) Line:219

Fig 3

4. The expression of miRNA in the tumors is mostly decreased compared to control tissue, but there is no significant change. No explanation is provided for the difference between plasma and tumor tissue.

We statistically evaluated mir-143-3p expression in colonic tumors.

Material and methods

Statistical analysis

(First version) No description

(Revised version) Line:160-161

The proportion of category was assessed by chi-square test.

Results

MiR-143-3p expression in colonic tumors

(First version) Line: 180-182

The expression of miR-143-3p in the colonic tumors was upregulated with fold change more than two in one patient (Table 3, Figure 2B). Nine patients exhibited less than half expression in normal mucosa compared to the matched tumor samples.

(Revised version) Line:214-217

The expression of miR-143-3p in the colonic tumors was upregulated with fold change more than two in one patient alone (Table 3). Nine patients exhibited less than half expression in normal mucosa compared to the matched tumor samples. Then, upregulation was significantly rare by chi-square test (P =0.02).

Discussion

(First version) Line: 189-191

In the current study, we explored circulating miRNAs specific for patients with FAP and found that miR-143-3p was upregulated in the plasma of patients with FAP, but not in colonic tumors.

(Revised version) Line:231-232

We found that miR-143-3p was upregulated in plasma, but not in colonic tumors. 

(First version) Line: 196-197

However, our current results suggested the upregulation of plasma miR-143-3p was insufficient for inhibiting tumor progression in patients with FAP.

(Revised version) Line:243-248

Rarity of miR-143-3p upregulation in colonic tumors of FAP patients is consistent with a previous report [26].

Then, our current results were inconsistent with the previous studies because upregulation of plasma miR-143-3p seemed insufficient for inhibiting tumor progression in FAP patients. Difference of miR-143-3p expression by sample location and its interaction has not been evaluated.

Overall, the major concern with this manuscript is the lack of rationale for the studies and the lack of description of the results.

Thank you for your comment. We revised introduction and discussion and added the data.

Our rationale is described in Introduction as below.

Introduction

Revised version, Line:71-77

Although most FAP cases are diagnosed by endoscopic examination without further genetic test, application of endoscopic examination for younger family is usually difficult to perform. Lynch syndrome, another hereditary disease, is difficult for diagnosis by endoscopic examination alone, and needs further examinations including genetic test [13]. Therefore, we chose FAP as a representative hereditary disease to evaluate the possibility of circulating miRNA as a potential biomarker of hereditary gastrointestinal diseases.

Description of the results were described as below.

Discussion

(First version) Line: 189-191

In the current study, we explored circulating miRNAs specific for patients with FAP and found that miR-143-3p was upregulated in the plasma of patients with FAP, but not in colonic tumors.

(First version) Line: 203-207

Circulating miRNAs are promising biomarkers for the diagnosis of CRC [21–23]. However, only a few reports have considered miR-143 as a biomarker of CRC [24–26]. On the other hand, miR-143 has been reported as a candidate biomarker for other diseases, including acute ischemic stroke, cerebral atherosclerosis, pituitary adenoma, and ulcerative colitis [27–30].

(Revised version) Line231-237

We found that miR-143-3p was upregulated in plasma, but not in colonic tumors. 

Circulating miRNAs are promising biomarkers for the diagnosis of CRC [15-17]. However, only a few reports have considered miR-143 as a biomarker of CRC [18-20]. On the other hand, circulating miR-143 has been upregulated in other diseases including acute ischemic stroke, cerebral atherosclerosis, amyotrophic lateral sclerosis, and pituitary adenoma [21-24].

Reference 24 has been changed from the manuscript about ulcerative colitis to amyotrophic sclerosis, because the former manuscript described regarding colonic tumors, but not blood samples.

(First version) Line:196-197

However, our current results suggested the upregulation of plasma miR-143-3p was insufficient for inhibiting tumor progression in patients with FAP.

(First version) Line:200-202

Our finding of the downregulation of miR-143 expression in colonic tumors of patients with FAP is consistent with a previous report [20].

(Revised version) Line243-248

Rarity of miR-143-3p upregulation in colonic tumors of FAP patients is consistent with a previous report [26].

Then, our current results were inconsistent with the previous studies because upregulation of plasma miR-143-3p seemed insufficient for inhibiting tumor progression in FAP patients. Difference of miR-143-3p expression by sample location and its interaction has not been evaluated.

Dear Reviewer 1 and Reviewer 2

Under reconstruction of the manuscript, we have changed the style of Abstract and remove one sample from analysis. Re-analysis has not changed significance of the data. 

1. We have also revised the abstract depending on the journal style.

2. We re-analyzed the data of colonic tumors, because the first version included the sample (F20) regarding SBA data, not colonic tumor. 

Abstract

(First version)

Background Circulating microRNAs (miRNAs) are considered promising biomarkers for diagnosis, prognosis, and treatment efficacy of malignancies. We assessed circulating miRNAs of patients with familial adenomatous polyposis (FAP) as a hereditary disease and prototype of colorectal cancer progression.

(Revised version)

Circulating microRNAs (miRNAs) are considered promising biomarkers for diagnosis, prognosis, and treatment efficacy of malignancies. We assessed circulating miRNAs of patients with familial adenomatous polyposis (FAP) as a hereditary disease and prototype of colorectal cancer progression to evaluate as possible biomarkers of hereditary gastrointestinal diseases.

(First version)

Methods MiRNAs specific for patients with FAP were identified using next-generation sequencing (NGS). Those that had moderate copy numbers and significant differences among the samples were selected for further evaluation as candidate biomarkers. With minimum differences among samples, miR-16-5p was selected as an internal control. The miRNAs were assessed by real-time PCR for expression ratios between minimum and maximum values at different time points. The miRNAs with the least expression variation were validated in patient plasma and colonic tumors.

Results MiR-143-3p, miR-96-5p, miR-183-5p, and miR-885-5p were selected as candidate biomarkers. Mean expression ratios of miR-143-3p/miR-16-5p, miR-143-3p/miR-103-3p, miR-183-5p/miR-16-5p, and miR-885-5p/miR-16-5p were 2.2, 3.4, 5.1, and 4.9, respectively. MiR-143-3p, with lowest variation ratio, was further assessed using specimens from 16 patients with FAP and 7 healthy donors. Plasma miR-143-3p was upregulated in FAP patients compared to healthy donors (P = 0.04), but not significantly influenced by sex, radical surgery, existence of desmoid tumor, or family history. However, miR-143-3p expression was not upregulated in colonic tumors of most patients with FAP, except for those with desmoid tumors.

Conclusions Our data suggested miR-143-3p expression differed between plasma and colonic tumors of most patients with FAP. Upregulation of plasma miR-143-3p expression may be helpful in the diagnosis of FAP, although it has no suppressive effect on tumor progression.

(Revised version)

Next-generation sequencing (NGS) indicated that plasma miR-143-3p, miR-183-5p, and miR-885-5p were candidate biomarkers for five FAP patients compared to three healthy donors due to moderate copy number and significant difference. MiR-16-5p was also assessed as an internal control due to minimum difference in expression across FAP patients and healthy donors. Then, plasma samples were further assessed by real-time PCR. Mean ratios of maximum expression and minimum expression were 2.2 for miR-143-3p/miR-16-5p, 3.4 for miR-143-3p/miR-103a-3p, 5.1 for miR-183-5p/miR-16-5p , and 4.9 for miR-885-5p/miR-16-5p by using eight FAP samples collected at different time points. Then, miR-143-3p/16-5p was further assessed using specimens from 16 FAP patients and 7 healthy donors. Plasma miR-143-3p was upregulated in FAP patients compared to healthy donors (P = 0.04), but not significantly influenced by pathological T factor, lymph node metastasis, sex, radical surgery, existence of desmoid tumor, or family history. However, miR-143-3p expression in colonic tumors was rare for upregulation, although there was a significant difference by existence of desmoid tumors. Our data suggested regulation of miR-143-3p expression differed by samples (plasma or colonic tumors) in most FAP patients. Upregulation of plasma miR-143-3p expression may be helpful for diagnosis of FAP, although suppressive effect on tumorigenesis seemed limited in FAP patients.

Results

MiR-143-3p expression in colonic tumors

(First version) Line-181-186

Nine patients exhibited less than half expression in normal mucosa compared to the matched tumor samples. There was no significant difference in miR-143-3p expression based on sex (P = 0.63), pathology of colonic tumor (P = 0.81), existence of gastric (P = 0.23) or duodenal polyp (P = 0.72), or family history (P = 0.36). However, there was a significant difference in miR-143-3p expression based on the existence or absence of desmoid tumor (P = 0.0002).

(Revised version) Line: 215-221

Eight patients exhibited less than half expression in normal mucosa compared to the matched tumor samples. Then, upregulation was significantly rare by chi-square test (P =0.02). There was no significant difference in miR-143-3p expression based on sex (P = 0.55), pathological T factor (P = 0.78, Fig. 3), existence of gastric (P = 0.14) or duodenal polyp (P = 0.59), or family history (P = 0.43). However, there was a significant difference in miR-143-3p expression based on the existence or absence of desmoid tumor (P <0.0001).

---

## [Decision Letter · Decision Letter 1]

3 Nov 2020

PONE-D-20-13212R1

Assessment of circulating microRNA specific for patients with familial adenomatous polyposis

PLOS ONE

Dear Dr. Yamano,

Thank you for submitting your manuscript to PLOS ONE. After careful consideration, we feel that it has merit but does not fully meet PLOS ONE’s publication criteria as it currently stands. Therefore, we invite you to submit a revised version of the manuscript that addresses the points raised during the review process.

Please address the comments provided by the reviewers. Especially reviewer #1 wants to see functionality via clinical samples or in vitro experiments. Final decision will be made based on the response (additional functional experiments).

We look forward to receiving your revised manuscript.

Kind regards,

Sripathi M Sureban, Ph.D.

Academic Editor

PLOS ONE

Additional Editor Comments (if provided):

Associate Editor:

Please address the comments provided by the reviewers. Especially reviewer #1 wants to see functionality via clinical samples or in vitro experiments. Final decision will be made based on the response (additional functional experiments).

Reviewer #1:

The authors have failed to address the comments raised about functionality. To simply cite other evidence and not generate any new results to address the queries is not enough. Also how is including "pathological T factor, lymph node metastasis" data association with miR of any help to understand miR specific disease function.

If the clinical samples were not available to perform the assays asked for the authors may want to use primary cells to address the experimental design.

Reviewer #2:

While the authors have addressed many of the reviewer comments, the rationale for this study is still lacking. Profiling for FAP patients would be done when? Would this happen after they have polyps? If that's the case, then based on age and other parameters, APC tests might be done. Lynch syndrome and FAP have different phenotypes, both of which are different than non-genetic CRC.

This has to be clarified well.

Reviewers' comments:

Reviewer's Responses to Questions

**Comments to the Author**

1. If the authors have adequately addressed your comments raised in a previous round of review and you feel that this manuscript is now acceptable for publication, you may indicate that here to bypass the “Comments to the Author” section, enter your conflict of interest statement in the “Confidential to Editor” section, and submit your "Accept" recommendation.

Reviewer #1: (No Response)

Reviewer #2: (No Response)

2. Is the manuscript technically sound, and do the data support the conclusions?

Reviewer #1: Partly

Reviewer #2: Partly

3. Has the statistical analysis been performed appropriately and rigorously? 

Reviewer #1: Yes

Reviewer #2: Yes

4. Have the authors made all data underlying the findings in their manuscript fully available?

Reviewer #1: Yes

Reviewer #2: Yes

5. Is the manuscript presented in an intelligible fashion and written in standard English?

Reviewer #1: No

Reviewer #2: No

6. Review Comments to the Author

Reviewer #1: (No Response)

Reviewer #2: While the authors have addressed many of the reviewer comments, the rationale for this study is still lacking. Profiling for FAP patients would be done when? Would this happen after they have polyps? If that's the case, then based on age and other parameters, APC tests might be done. Lynch syndrome and FAP have different phenotypes, both of which are different than non-genetic CRC.

This has to be clarified well.

7. PLOS authors have the option to publish the peer review history of their article (what does this mean?). If published, this will include your full peer review and any attached files.

Reviewer #1: No

Reviewer #2: No

---

## [Author Response · Author response to Decision Letter 1]

16 Feb 2021

Dear Reviewers,

Thank you for your time for reviewing the manuscript and describing the useful comments to us again. We have revised the manuscript and responded to your comments as possible as we could.

Associate Editor:

Please address the comments provided by the reviewers. Especially reviewer #1 wants to see functionality via clinical samples or in vitro experiments. Final decision will be made based on the response (additional functional experiments).

Thank you for giving us to revise the manuscript again. We have performed in vitro experiments depending on Reviewer 1 suggestion and also revised the manuscript depending on Reviewer 2 suggestions. 

Reviewer #1:

The authors have failed to address the comments raised about functionality. To simply cite other evidence and not generate any new results to address the queries is not enough. Also how is including "pathological T factor, lymph node metastasis" data association with miR of any help to understand miR specific disease function.

If the clinical samples were not available to perform the assays asked for the authors may want to use primary cells to address the experimental design.

Thank you for your comments. We have performed functional experiments to respond your comments using colorectal cancer cell lines, Caco2, HCT116, LoVo, and RKO instead of clinical samples. In these experiments, we assessed effect of mir-143-3p transfection on proliferation of these cells together with transfection reagent alone and transfection with control microRNA. Although there are several reports regarding effect of mir-143-3p on proliferation of colorectal cancer cells, the methods and results were different among the reports. Our results were most similar with that reported by Luo et al, which was added in References No. 32).

We added the sentences below in Abstract, Material and methods, Results, and Discussion.

Abstract, Line 46-47

MiR-143-3p transfection significantly inhibited colorectal cancer cell proliferation compared to control microRNA transfection.

Introduction, Line 83-84

We also assessed the effect of the candidate miRNA on proliferation of colorectal cancer cells.

Material and methods, Line 165-189

Cell culture

Human CRC cell lines, Caco2, HCT116, LoVo, and RKO were kind gifts from Dr. Hirofumi Yamamoto (Osaka University Graduate School of Medicine, Osaka, Japan ). These cells were authenticated by ATCC service. These cells except for RKO were cultured by Roswell Park Memorial Institute 1640 medium with 10% fetal bovine serum (FBS). RKO was cultured by Dulbecco’s modified Eagle’s medium with FBS. Culture media and FBS was purchased from Nacalai Tesque (Kyoto, Japan) and Atlas Biologicals (CO, USA) , respectively. Cell experiments were maintained at 37℃ in a humidified atmosphere of 5% CO2. 

Cell transfection 

Cells were seeded in 96-well flat-bottom plate at a density of 5000 cells/well for Caco-2, 500 cells/well for HCT116, 1000 cells/well for LoVo, and 500 cells/well for RKO, respectively. 24 hours after seeding, cells were treated with Lipofectamine 2000 and Opti-MEM containing 5 pmol of miRNA Mimic Negative Control (Control treatment), 5 pmol of hsa-miR-143-3p (miR-143 treatment), and no miRNA (Mock treatment) according to the manufacturer’s instructions. Transfection reagents including miRNAs were purchased from Thermo Fisher Scientific. 

Cell proliferation assay 

In vitro growth was evaluated using alamarBlue Cell Viability Reagent at one, two, three, and four days after transfection. The ration of the fluorescence intensity at 590 nm by treatment groups compared to Mock treatment was calculated. Effect of miR-143-3p on cell growth was evaluated by comparison of the ratio between miR-143 treatment and Control treatment using Student’s t-test.

Results, Line 267-283

Effect of miR-143-3p on the proliferation of colorectal cancer cells

　We assessed the effect of miR-143-3p on the proliferation of CRC cell lines (Caco2, HCT116, LoVo, and RKO). Except for Caco2, miR-143 treatment showed significant growth inhibition on these cell lines compared to Control treatment at four days after transfection (p < 0.01, Fig. 4A, B, C, D). MiR-143 treatment also showed significant inhibition of HCT116 and LoVo compared to Control treatment at three days after transfection (p < 0.01, Fig. 4B and 4C). In Caco2, effect of miR-143-3p on proliferation was not significant at four days after transfection (Fig. 4A, P = 0.06).

Fig. 4. The effect of miR-143-3p on the proliferation of colorectal cancer cells.

The effect of miR-143-3p on the cell growth was evaluated at one, two, three, and four days after transfection. There was no significant effect on proliferation of Caco2 (A). MiR-143-3p showed significant effect on proliferation of HCT116 (B) and LoVo (C) at three and four days after transfection (p < 0.01). MiR-143-3p showed significant effect on proliferation of RKO (D) at four days after transfection (p < 0.01). Results are expressed as mean of the ratio of cell proliferation compared to transfection without miRNA ± standard deviation. * p <0.01 compared to Control treatment.

Discussion, Line 296-299

Our data showed that transfection of miR-143-3p inhibited proliferation of CRC cell lines, although the extent of growth inhibition differed among these cell lines. Our results are similar to the previous reports, especially by Luo et al [26,30,32]. 

Discussion, Line 306-308

However, transfection of miR-143-3p should be more effective on cell proliferation than plasma miR-143-3p. 

Discussion, Line 320-321

First, biological effect of miR-143-3p have not been evaluated by using colonic samples derived from FAP patients or APC-deficient mice, although we have shown inhibitory effect of miR-143-3p on proliferation by using CRC cell lines. 

Reviewer #2:

While the authors have addressed many of the reviewer comments, the rationale for this study is still lacking.

Thank you for your comment. Our rationale is described in Introduction as below.

“Circulating miRNAs are recognized as diagnostic or prognostic biomarkers of many diseases [3]; however, their usefulness in the diagnosis of gastrointestinal hereditary diseases has not been reported [4–6]. Genetic variants in patients with hereditary disease may induce consistent and specific miRNA profile differences in their blood compared to healthy donors.”

“However, we hypothesized that plasma miRNA profile of FAP patients, which bears APC variants in all cells, may differ from that of patients with sporadic CRC and healthy donors.”

However, these sentences could not introduce the readers what we have done in this study as your comments. We revised Abstract and added the sentences below in Introduction.

Abstract

(Previous version) Line 28-31

Circulating microRNAs (miRNAs) are considered promising biomarkers for diagnosis, prognosis, and treatment efficacy of malignancies. We assessed circulating miRNAs of patients with familial adenomatous polyposis (FAP) as a hereditary disease and prototype of colorectal cancer progression to evaluate as possible biomarkers of hereditary gastrointestinal diseases.

(Revised version) Line 30-33

Circulating microRNAs (miRNAs) are considered promising biomarkers for diagnosis, prognosis, and treatment efficacy of diseases. However, usefulness of circulating miRNAs as biomarkers for hereditary gastrointestinal diseases have not been confirmed yet. We explored circulating miRNAs specific for patients with familial adenomatous polyposis (FAP) as a representative hereditary gastrointestinal disease.

Introduction

(Revised version) Line 81-84

In this study, we explored the candidate plasma mir-RNAs specific for FAP patients by Next-generation sequencing (NGS) at first and validated the candidate miRNA by further experiments using clinical samples. We also assessed the effect of the candidate miRNA on proliferation of colorectal cancer cells. 

Profiling for FAP patients would be done when? Would this happen after they have polyps? If that's the case, then based on age and other parameters, APC tests might be done. Lynch syndrome and FAP have different phenotypes, both of which are different than non-genetic CRC. This has to be clarified well.

Thank you for comments regarding the diagnosis of FAP. We diagnosed FAP by clinical features, not by genetic test, because cost of genetic test is not supported by insurance in Japan. Then, the patients are charged several thousand dollars for genetic test. Then, the patients usually do not request for genetic test. Endoscopic findings of stomach (gastric polyp) and duodenum (duodenal polyp) are helpful for distinguish between FAP and Lynch. Different from FAP, genetic test for Lynch is supported by insurance in Japan.

In this study, 16 of 18 patients who had gastric polyps and/or duodenal polyps could be diagnosed as FAP, not Lynch, without genetic test. 

We have revised the sentences as below.

Material and methods

(Previous version) Line 83-86

Ethylenediaminetetraacetic acid containing plasma samples were collected from 18 patients clinically diagnosed endoscopically with FAP. The patients included those with and without a family history of FAP and were treated at our hospital between 2015 and 2018.

(Re-revised version) Line 90-95

Ethylenediaminetetraacetic acid containing plasma samples were collected from 18 patients who were diagnosed as FAP by clinical features including family history and endoscopic findings in not only colon and rectum but also stomach and duodenum. Genetic tests for diagnosis of FAP have not been performed in this study, because these tests were not covered by insurance in Japan. These patients were treated at our hospital between 2015 and 2018.

---

## [Decision Letter · Decision Letter 2]

31 Mar 2021

Assessment of circulating microRNA specific for patients with familial adenomatous polyposis

PONE-D-20-13212R2

Dear Dr. Yamano,

We’re pleased to inform you that your manuscript has been judged scientifically suitable for publication and will be formally accepted for publication once it meets all outstanding technical requirements.

Kind regards,

Sripathi M Sureban, Ph.D.

Academic Editor

PLOS ONE

Additional Editor Comments (optional):

Great work on the role of miR-143 - Congratulations!

Reviewers' comments:

Reviewer's Responses to Questions

**Comments to the Author**

1. If the authors have adequately addressed your comments raised in a previous round of review and you feel that this manuscript is now acceptable for publication, you may indicate that here to bypass the “Comments to the Author” section, enter your conflict of interest statement in the “Confidential to Editor” section, and submit your "Accept" recommendation.

Reviewer #1: All comments have been addressed

Reviewer #2: All comments have been addressed

2. Is the manuscript technically sound, and do the data support the conclusions?

Reviewer #1: Yes

Reviewer #2: (No Response)

3. Has the statistical analysis been performed appropriately and rigorously? 

Reviewer #1: Yes

Reviewer #2: (No Response)

4. Have the authors made all data underlying the findings in their manuscript fully available?

Reviewer #1: Yes

Reviewer #2: (No Response)

5. Is the manuscript presented in an intelligible fashion and written in standard English?

Reviewer #1: Yes

Reviewer #2: (No Response)

6. Review Comments to the Author

Reviewer #1: (No Response)

Reviewer #2: (No Response)

7. PLOS authors have the option to publish the peer review history of their article (what does this mean?). If published, this will include your full peer review and any attached files.

Reviewer #1: **Yes: **RISHEIN GUPTA

Reviewer #2: No

---

## [Editor Report · Acceptance letter]

23 Apr 2021

PONE-D-20-13212R2 

Assessment of circulating microRNA specific for patients with familial adenomatous polyposis 

Dear Dr. Yamano:

I'm pleased to inform you that your manuscript has been deemed suitable for publication in PLOS ONE. Congratulations! Your manuscript is now with our production department. 

Kind regards, 

on behalf of

Dr. Sripathi M Sureban 

Academic Editor

PLOS ONE